# Self-Medication during the COVID-19 Pandemic in Brazil: Findings and Implications to Promote the Rational Use of Medicines

**DOI:** 10.3390/ijerph20126143

**Published:** 2023-06-16

**Authors:** Patrícia Silva Bazoni, Ronaldo José Faria, Francisca Janiclecia Rezende Cordeiro, Élida da Silva Timóteo, Alciellen Mendes da Silva, Ana Luisa Horsth, Eduardo Frizzera Meira, Jéssica Barreto Ribeiro dos Santos, Michael Ruberson Ribeiro da Silva

**Affiliations:** 1Pharmaceutical Services Graduate Program, Federal University of Espírito Santo, Alegre 29500-000, Brazil; patricia.bazoni@yahoo.com.br (P.S.B.); ronaldojfaria@hotmail.com (R.J.F.); 2Health Technology Assessment and Economy Group, Federal University of Espírito Santo, Alegre 29500-000, Brazil; franrcordeiro1@gmail.com (F.J.R.C.); elidatimoteo8@gmail.com (É.d.S.T.); alciellen.silva@edu.ufes.br (A.M.d.S.); ana.horsth@edu.ufes.br (A.L.H.); eduardo.meira@ufes.br (E.F.M.); jessica.br.santos@ufes.br (J.B.R.d.S.)

**Keywords:** self-medication, prevalence, cross-sectional study, pharmacoepidemiology

## Abstract

Self-medication is identified by the consumption of medications without a prescription or guidance from a qualified prescribing professional. This study estimated the prevalence, profile, and associated factors with self-medication during the COVID-19 pandemic in Brazil. A cross-sectional study was conducted through a household survey in the Alegre city, from November to December 2021. Descriptive analysis was performed for the sociodemographic and clinical characteristics of the interviewees. Poisson regression with robust variance was used to identify the association of sociodemographic and clinical variables with self-medication. A total of 654 people were interviewed, of whom 69.4% were self-medicating. The younger age group (PR = 1.13; 95% CI = 1.01–1.26), female gender (PR = 1.19; 95% CI = 1.04–1.37), consumption of alcoholic beverages (PR = 1.13; 95% CI = 1.01–1.25), and problems with adherence to pharmacological treatment (PR = 1.15; 95% CI = 1.04–1.28) were associated with self-medication, while the occurrence of polypharmacy (PR = 0.80; 95% CI = 0.68–0.95) was a protective factor for self-medication. Self-medication was directly related to over-the-counter drugs, with analgesics dipyrone and paracetamol being the most commonly used. Self-medication consumption of prescription drugs, including those under special control, was identified to a lesser extent.

## 1. Introduction

Medicines are considered one of the main resources for maintaining health as they positively help in improving the quality of life of the population [1]. The drug treatment must be used correctly to achieve effectiveness, safety, and success, as it can damage to the patient’s health when used improperly [2,3].

Self-medication stands out among the factors that may contribute to the emergence of problems related to the use of medicines [4]. It is an escalating global phenomenon that poses a public health challenge, primarily due to concerns such as antibiotic resistance, the potential for harmful side effects, drug interactions, and the possibility of masking underlying diseases [5]. Based on several consensuses in the literature, self-medication can be understood as the consumption of medicines without the prescription of qualified healthcare professionals, the reuse of drugs that were previously prescribed, or the modification of their use [6].

Many individuals see self-medication as self-care since it is considered to be a practice that brings benefits to society and can be seen as complementary to healthcare services, causing a reduction in overcrowding in healthcare units, and providing greater availability of the services offered. Furthermore, this practice can save time and money for both users and healthcare services [6].

Political, cultural, and economic factors have contributed to the increase in self-medication, making it one of the main public health problems [7]. The ease of access to medicines, as well as their promotion and advertising in the pharmaceutical market, can influence the practice of self-medication, which contributes to the unnecessary and inappropriate use of these products [8]. Moreover, it may be influenced by the population’s difficulty in accessing public healthcare services and affording a private health insurance plan [9,10].

Another factor that strongly contributes to self-medication is the existence of over-the-counter drugs (OTCs) on the market. They are marketed in the auto services of drugstores and pharmacies, and presenting a medical prescription is not mandatory to buy them [11].

Furthermore, the COVID-19 pandemic brought a new challenge for the responsible use of medicines, which has been called an infodemic. The term infodemic is associated with the excessive sharing of inhomogeneously accurate information in response to an acute situation such as the COVID-19 pandemic, amplified by the efficient and multiple means of dissemination and collective fear [12].

The practice of self-medication can cause significant damage to the user’s life [13]. The lack of knowledge about the indications and dosages of medicines, including easy access to them, and the lack of information leads to several situations that can cause serious damage and consequences to the user’s health. These situations may include incorrect administration of drugs, inadequate dosage, incorrect route of drug administration, insufficient or a longer-than-necessary treatment time, incorrect self-diagnosis, errors in the choice of drug therapy, masking of serious diseases, the emergence of adverse events, risk of dependence and abuse, among others [8,10].

Therefore, this study aims to estimate the prevalence, profile, and factors associated with the occurrence of self-medication during the COVID-19 pandemic in residents of a municipality in the southern region of Espírito Santo. Thus, it will be possible to determine how self-medication occurs in the municipality to propose changes in public health policy, as well as health education efforts for the population.

## 2. Materials and Methods

An epidemiological study was conducted with a cross-sectional design through a household survey in the municipality of Alegre from November to December 2021. This municipality is located in the southern region of the state of Espírito Santo, with an estimated population of 29,869 inhabitants, in 2021, distributed between the headquarters of the municipality and seven districts: Anutiba, Araraí, Café, Celina, Rive, Santa Angélica, and São João do Norte [14].

### 2.1. Study Population and Sampling

The study population consisted of individuals living in the headquarters and districts of the municipality of Alegre, aged ≥ 18 years, who agreed to participate in the research by signing an informed consent form.

The sample size was estimated considering reference to the urban population of 21,512, at a confidence level of 95% (error α = 0.05), the estimated prevalence of 50% for different prevalence endpoints of the study, and the design effect of 1.5. Based on these parameters, the final minimum sample was estimated at 567 individuals, to which 10% was added to cover possible losses, totaling 624 individuals to be interviewed [15].

For the identification of the individuals to be interviewed, sampling with probabilities proportional to size was used, according to the methodology described by the WHO [16,17]. In the first stage, 10 out of 37 urban census tracts of Alegre were randomly selected. In the second stage, a similar number of individuals should be interviewed in each sector.

### 2.2. Data Collection

Data collection was performed through an interview in which individuals were asked the relevant questions about the sociodemographic data, general health, COVID-19, use of health services, use of medicines, use of medicinal plants, lifestyle habits, and quality of life. A structured and pre-coded questionnaire was used to collect the data, and all responses were self-reported by the individuals.

Data collection occurred during the daytime hours for three consecutive weeks in November and December 2021 to minimize researchers’ exposure during the COVID-19 Pandemic. All researchers adhered to the safety protocols implemented in Brazil, including hand hygiene with alcohol, wearing face masks, and using personal protective equipment.

Before starting the fieldwork, practical training of the researchers was conducted, in which information about the data collection instrument and the process of working in the field was reinforced. Additionally, a pilot study was conducted for the test and evaluation of the questionnaire, as well as for the practical training of researchers. The technical team responsible for training consisted of professors and researchers from the *Grupo de Avaliação*, *Tecnologia e Economia em Saúde* (Health Technology Assessment and Economy Group—GATES) of the Federal University of Espírito Santo (UFES).

### 2.3. Study Variables

The dependent variable was the occurrence of self-medication, defined as the use of medicines on its own to treat health problems, obtained from the following question: “Do you usually take any medicine on your own?”.

The independent variables were age, gender, skin color, region, marital status, religion, schooling level, income, self-perception of health, quality of life measured by the EuroQoL 5 Dimensions 3 levels instrument (EQ-5D-3L), body mass index (BMI) classification, regular practice of physical activity, consumption of alcoholic beverages, smoking, hours of daily sleep, medical and dental consultations, consultations with a nutritionist in the last year, hospitalizations in the last year, coverage by a private health insurance plan, use of the *Farmácia Básica Municipal* (municipal primary pharmacy) and private pharmacies in the last year, problems of treatment, and use of medicinal plants.

The drugs most used by the population in self-medication were described and organized according to the following classifications: (a) If the drug was available in the Brazilian National List of Essential Medicines (RENAME), which is part of the Unified Health System (SUS), the public health system of Brazil; (b) If the drug was accessible through pharmaceutical services within SUS, categorized as either Basic (for primary healthcare drugs), Strategic (for drugs used in endemic and epidemiological situations in Brazil), or Specialized (for high-cost drugs); (c) if the drug was available when prescribed, available over-the-counter, or subject to special control (restricted drugs, such as psychotropic). 

### 2.4. Data Analysis

Descriptive analysis was performed by frequency distribution for categorical variables and by median and interquartile interval for continuous variables.

The factors associated with self-medication were analyzed by bivariate and multivariate analysis. The variables that indicated *p* ≤ 0.20 for association with the dependent variable in the bivariate analysis were included in the multivariate analysis, using the Poisson regression model with robust variance. Only the variables associated with the *p*-value ≤ 0.05 remained in the final model. All data were analyzed using the Jamovi 2.2.5 software, except for the Poisson regression, which was performed in Stata 16.1.

### 2.5. Ethical Considerations

This study was approved by the Research Ethics Committee of the Federal University of Espírito Santo (UFES), *Alegre campus*, under opinion No. 3428060. All interviewees were informed about the research and agreed to participate by signing of the informed consent form, and were guaranteed the confidentiality and anonymity of the information obtained.

## 3. Results

A total of 654 people were interviewed, of whom 73.5% (*n* = 481) were female, 45.5% (*n* = 295) aged ≥ 60, with the median age of 56 years (interquartile range (IQR) of 37.0–67.0). Among the interviewees who reported self-medication, 76.9% (*n* = 349) were female, 59.6% (*n* = 269) were aged under 60, and 70.9% (*n* = 321) lived in the headquarters of the municipality of Alegre. Regarding the skin color variable, 48.7% (*n* = 220) declared themselves white. A total of 42.8% (*n* = 194) were married, and 49.3% (*n* = 224) were Catholic. When asked about their schooling level, 36.1% (*n* = 164) answered that they had incomplete elementary school; and concerning income, 48.5% (*n* = 207) of the interviewees said that they received up to one per capita minimum wage (Table 1).

Regarding the clinical and health characteristics of the interviewed population, self-medication was present in 50.9% (*n* = 231) of the individuals who reported having a very good or good self-perception of health. Furthermore, the median quality of life estimated by the EQ-5D-3L was 0.884. Regarding the body mass index of the interviewees, 38.8% (*n* = 167) were overweight, and 29.3% (*n* = 126) were obese. When asked about the practice of regular physical activity, 65.7% (*n* = 297) did not practice it, 27.2% (*n* = 123) consumed alcoholic beverages, and 13.0% (*n* = 59) smoked. Regarding the hours of daily sleep, 34.4% (*n* = 156) answered that they slept from 7 to 8 h a day. Moreover, in the last year, 20.1% (*n* = 364) underwent medical consultations, 35.9% (*n* = 172) underwent dental consultations, 9.7% (*n* = 41) consulted a nutritionist, and 11.3% (*n* = 51) were hospitalized. A total of 76.4% (*n* = 347) of the interviewees had no private health plan. In the last year, 53.0% (*n* = 239) of the interviewees used the municipal primary pharmacy and 89.6% (*n* = 407) used some private pharmacy (Table 2).

Additionally, 69.4% (*n* = 454) of the interviewees reported using medications on their own (self-medication), 49.6% (*n* = 225) were in minor polypharmacy (using two to four drugs), 18.3% (*n* = 83) were in major polypharmacy (using five or more drugs), 28.8% (*n* = 128) had problems with pharmacotherapy, and 41.2% (*n* = 183) used medicinal plants (Table 2).

The Poisson regression analysis indicated that the factors associated positively and significantly with the occurrence of self-medication were the lowest age group, women, patients who consume alcoholic beverages, those who presented problems of adhering to pharmacological treatment, and, as a protective factor, those who were in major polypharmacy (Table 3).

Regarding age, there was a higher prevalence of self-medication among individuals who were aged under 60 (prevalence ratio (PR) = 1.13; 95% CI = 1.01–1.26), in women (PR = 1.19; 95% CI = 1.04–1.37), in those who consume alcoholic beverages (PR = 1.13; 95% CI = 1.01–1.25), and those who presented treatment adherence problems (PR = 1.15; 95% CI = 1.04–1.28). Additionally, a lower prevalence of self-medication was observed in individuals who were in major polypharmacy (PR = 0.80; 95% CI = 0.68–0.95). Statistical significance was not observed in the analysis of any of the other factors examined (*p* > 0.05) (Table 3).

Table 4 shows the medicines most used by the population that practices self-medication in the municipality of Alegre, Espírito Santo. Dipyrone was reported by 35.7% of the interviewees, followed by paracetamol (9.5%). The most used drugs were over-the-counter (OTC). However, the partial use of levonorgestrel associated with ethinylestradiol (five interviewees) and omeprazole (two) was identified by self-medication, and these drugs were subject to medical prescription. Finally, the use of controlled drugs by self-medication was identified, such as fluoxetine (one), alprazolam (one), bromazepam (one), and citalopram (one). Overall, 223 different drugs in use were reported. Losartan, hydrochlorothiazide, and simvastatin were the medicines most used by the population that did not practice self-medication (Table 5).

Most of these drugs were incorporated into the Brazilian National List of Medicines (RENAME), in addition to belonging to the Basic Component of Pharmaceutical Care (CBAF) used in primary care. Following the Anatomical Therapeutic Chemical Classification System (ATC), the anatomical groups most consumed by the population of Alegre were those of the cardiovascular system (C), followed by the central nervous system (N), and Food Treatment and Metabolism (A) (Table 4 and Table 5).

## 4. Discussion

Self-medication is a very common practice that can lead to delayed diagnosis and treatment of diseases [10]. This practice poses a significant global public health challenge, raising concerns about antibiotic resistance, potential adverse effects, drug interactions, and the masking of underlying diseases [5]. In this study, a high proportion of individuals who practice self-medication was observed. Out of the 654 individuals interviewed through the household survey, 454 (69.4%) self-medicated, i.e., use medicines on their own to treat self-recognized health problems.

These data are similar to those of a cross-sectional study conducted in the municipality of Crato, state of Ceará, from 2013 to 2014, in which 104 individuals were interviewed. The authors observed a prevalence of self-medication of 67.65% [8]. Another study conducted in Brazil, involving 270 individuals (181 adolescents and 89 adults), with a mean age of 23.1 ± 10.8, showed that 69.3% of them practiced self-medication [18].

Studies in several regions and countries show that the prevalence of self-medication may vary according to the population studied, as shown by a survey conducted by Wirowski et al. [19], from July to September 2021, to verify the prevalence of self-medication among young adults (18 to 35 years) during the COVID-19 pandemic in Brazil. They interviewed 349 individuals and observed a prevalence of self-medication of 32.7% (*n* = 114).

However, a survey conducted by the Brazilian Federal Pharmacy Council (CFF), through the Datafolha Institute in 2019, verified the occurrence of self-medication in 77% of the Brazilian population that used medicines in the last six months. Moreover, 47% of the interviewees self-medicated at least once a month and 25% of them practiced self-medication every day or at least once a week [20].

Similar to Brazil, studies in other countries show a variation in the occurrence of self-medication, as in an African cross-sectional study conducted with 609 clients from a pharmacy in Asmara, Eritrea, which identified the prevalence of self-medication with over-the-counter drugs in 93.7% of the respondents [21].

Amaral et al. [13] conducted a study with residents of Central and Northern Portugal, in which 197 individuals were interviewed. They observed the prevalence of self-medication in 74.1% of the interviewees throughout their lives, and in the last six months, this prevalence was 59.9%. Thus, the prevalence of self-medication may vary depending on the population and region in which the study is conducted.

The differences in self-medication practices are likely attributed to various factors that drive individuals to self-medicate. These factors include a lower perceived severity of the illness, limited time to visit a healthcare professional, easy access to medications, previous positive experiences with self-medication, and the high costs associated with seeking professional healthcare services [22].

This study identified various factors associated with self-medication. Risk factors included a younger age (<60 years), female gender, alcohol consumption, and difficulties in adhering to the prescribed treatments. Conversely, the use of five or more medications (polypharmacy) was found to be a protective factor. Other studies found additional factors associated with self-medication such as drug use influenced by advertising, drug suggestions from nonhealthcare professionals, self-reported good health, long time since last medical consultation [18], older age, presence of chronic diseases, difficulties in daily activities [23], and sociocultural influences, including media advertisements and financial limitations in accessing medical care [24]. Furthermore, factors such as a lower education level, religion, knowledge about over-the-counter drugs [21], living in urban areas, proximity to healthcare units, better economic conditions, and higher education level were associated with self-medication [25].

In this study, the data collection was conducted during the pandemic period of COVID-19, which may have corroborated the increase in self-prescribed medication use among the interviewees. In this pandemic period, a transition from daily activities to the online format occurred, through online classes and home-office work, which may have contributed to an increase in health risks for the population. Inadequate ergonomics and the extensive hours individuals spend in front of electronic devices are factors that may be associated with musculoskeletal pain and headaches [26,27].

Among the most-used drugs by self-medication in this study are analgesics, antipyretics, muscle relaxants, anti-inflammatories, and anti-flu drugs. These findings are similar to those of a study conducted by Gonçalves Júnior et al. [8], who found that the most commonly self-medicated drugs consumed by the population studied were analgesics, antipyretics, anti-flu medications, drugs for gastric discomfort, and antibiotics.

During the COVID-19 pandemic, there was an increase in the consumption of over-the-counter (OTC) drugs, especially analgesics, anti-inflammatories, and muscle relaxants. These medications were used to treat COVID-19-related symptoms such as fever, headache, and muscle pain. Analgesics are commonly used for self-medication to alleviate headaches and muscular pain [19,28,29,30,31]. The ease of acquisition in pharmacies and drugstores of OTC drugs, without the need for a medical prescription, likely contributes to the widespread use of these drug classes [32]. In this regard, it is important to implement measures such as pharmaceutical counseling and health education for patients who practice self-medication to ensure the safe and informed use of OTC drugs. These measures aim to prevent the occurrence of adverse events that may result from insufficient knowledge regarding the appropriate usage of these medications [31,33].

This study had limitations that should be considered. Cross-sectional studies do not allow inferring causality, since they do not consider the time variable in their analysis; however, they provide relevant information that can guide longitudinal studies. As a recall period of fifteen days was used to evaluate the use of medications, a memory bias may have occurred. To minimize the occurrence of this bias, the patient’s report of drug usage was proven by providing either the medical prescription or the original packaging of the drugs being used. Finally, we could not detail the reasons related to self-medication, such as the use of medication for maintenance of treatment from expired prescriptions. 

There are some concerns regarding the generalization of the results of this study. First, data collection occurred between 8 a.m. and 6 p.m., when many people were at their workplaces and away from their residences. This may have resulted in selection bias, favoring the participation of older individuals and females, which may have overestimated the prevalence of self-medication since these factors were associated with the presence of this practice. Additionally, the study was conducted in a single municipality in southeastern Brazil, which did not capture the full geographic, economic, social, and cultural diversity found in the country. Despite these concerns, the study is relevant as it addresses the topic of self-medication during the COVID-19 pandemic, making the article original and timely. Furthermore, it provides novel and representative data from the region in which it was conducted, enabling the planning and organization of health actions and services to improve the population’s quality of life and welfare.

## 5. Conclusions

The high proportion of self-medicating individuals in this study reveals the importance of awareness of the risks of self-medication. Another study can be developed based on this one to comprehend the main reasons for self-medication in the population. Furthermore, actions to raise awareness of this population are important, to the extent that they contribute to the formation of citizens aware of their share of responsibility for the responsible use of medicines. Additionally, the results of this study may contribute to the improvement of patient care, as well as provide subsidies for public health promotion policies.

## Figures and Tables

**Table 1 ijerph-20-06143-t001:** Sociodemographic characteristics of the population of Alegre, Espírito Santo.

Variables	No Self-Medication	Self-Medication	*p*-Value	Total
***Age in years*** *(median*, *IQR)*	63.0 (46.0–71.0)	54.0 (35.0–67.0)	<0.001	56.0 (37.0–67.0)
***Age group*** *(n*, *%)*			<0.001	
<60 years	85 (42.9)	269 (59.6)		354 (54.5)
≥60 years	113 (57.1)	182 (40.4)		295 (45.5)
***Gender*** *(n*, *%)*			0.004	
Women	132 (66.0)	349 (76.9)		481 (73.5)
Men	68 (34.0)	105 (23.1)		173 (26.5)
***Skin color*** *(n*, *%)*			0.876	
White	93 (46.5)	220 (48.7)		313 (48.0)
Mixed-race	69 (34.5)	149 (33.0)		218 (33.4)
Other	38 (19.0)	83 (18.4)		121 (18.6)
***Region*** *(n*, *%)*			0.322	
Headquarters	134 (67.0)	321 (70.9)		455 (69.7)
District	66 (33.0)	132 (21.1)		198 (30.3)
***Marital status*** *(n*, *%)*			0.844	
Single	48 (24.0)	117 (25,8)		165 (25.3)
Married	90 (45.0)	194 (42.8)		284 (43.5)
Other	62 (31.0)	142 (31.3)		204 (31.2)
***Religion*** *(n*, *%)*			0.13	
No religion	79 (3.5)	39 (8.6)		46 (7.0)
Catholic	106 (53.0)	224 (49.3)		330 (50.5)
Protestant	73 (36.5)	163 (35.9)		236 (36.1)
Other	14 (7.0)	28 (6.2)		42 (6.4)
***Schooling level*** *(n*, *%)*			0.009	
Incomplete primary education	96 (48.0)	164 (36.1)		260 (39.8)
Completed high school	87 (43.5)	227 (50.0)		314 (48.0)
Completed technician or higher education	17 (8.5)	63 (13.9)		80 (12.2)
***Per capita income*** *(n*, *%)*				
Up to 1 minimum wage	78 (40.6)	207 (48.5)	0.166	285 (46.0)
From 1 to 2 minimum wages	93 (48.4)	174 (40.7)		267 (43.1)
More than 2 minimum wages	21 (10.9)	46 (10.8)		10 (10.8)

BMI: Body mass index (BMI); IQR: Interquartile range; n: number of interviewees per variable concerning the total number of interviewees; %: percentage of the variable about the total number of interviewees. Note: Missing data were not considered in the analyses.

**Table 2 ijerph-20-06143-t002:** Clinical characteristics of the population of Alegre, Espírito Santo.

Variables	No Self-Medication	Self-Medication	*p*-Value	Total
***Quality of life*** *(median*, *IQR)*	0.881 (0.751–1.000)	0.884 (0.817–1.000)	0.213	0.884 (0.817–1.000)
***Self-perceived health*** *(n*, *%)*			0.162	
Very good/good	99 (49.5)	231 (50.9)		330 (50.5)
Regular	78 (39.0)	191 (42.1)		269 (41.1)
Bad/very bad	23 (11.5)	32 (7.0)		55 (8.4)
***BMI classification*** *(n*, *%)*			0.279	
Normal weight	7 (3.9)	15 (3.5)		22 (3.6)
Underweight	65 (35.9)	122 (28.4)		187 (30.6)
Overweight	65 (35.9)	167 (38.8)		232 (38.0)
Obesity	44 (24.3)	126 (29.3)		170 (27.8)
***Regularly performs physical activity*** *(n*, *%)*			0.445	
Yes	75 (37.7)	157 (34.6)		232 (35.5)
No	124 (62.3)	297 (65.4)		421 (64.5)
***Consumption of alcoholic beverages*** *(n*, *%)*			0.009	
Yes	35 (17.6)	123 (27.2)		158 (24.2)
No	164 (82.4)	330 (72.8)		494 (75.8)
***Smoker*** *(n*, *%)*			0.587	
Yes	29 (14.6)	59 (13.0)		88 (13.5)
No	170 (85.4)	395 (87.0)		565 (8)
***Hours of daily sleep*** *(n*, *%)*			0.089	
<6 h	42 (21.2)	111 (24.5)		153 (23.5)
From 6 to 7 h	61 (30.8)	117 (25,8)		178 (27.3)
From 7 to 8 h	54 (27.3)	156 (34.4)		210 (32.3)
>8 h	41 (20.7)	69 (15.2)		110 (16.9)
***Medical consultations in the last year*** *(n*, *%)*			0.852	
Yes	159 (79.9)	364 (20.1)		523 (80.3)
No	40 (20.1)	88 (19.5)		128 (19.7)
***Dental consultations in the last year*** *(n*, *%)*			0.137	
Yes	65 (33.3)	172 (35.9)		237 (37.6)
No	130 (66.7)	263 (60.5)		393 (62.4)
***Consultations with a nutritionist in the last year*** *(n*, *%)*			0.734	
Yes	17 (8.8)	41 (9.7)		58 (9.4)
No	176 (91.2)	383 (90.3)		559 (90.6)
***Hospitalizations in the last year*** *(n*, *%)*			0.181	
Yes	30 (15.0)	51 (11.3)		81 (12.4)
No	170 (85.0)	402 (88.7)		572 (87.6)
***Private health insurance plan*** *(n*, *%)*			0.874	
Yes	46 (23.0)	107 (23.6)		153 (23.4)
No	154 (77.0)	347 (76.4)		501 (76.6)
***Used municipal primary pharmacy in the last year*** *(n*, *%)*			0.280	
Yes	114 (57.6)	239 (53.0)		353 (54.4)
No	84 (42.4)	212 (47.0)		296 (45.6)
***Used private pharmacy in the last year*** *(n*, *%)*			0.233	
Yes	172 (86.4)	407 (89.6)		579 (88.7)
No	27 (13.6)	47 (10.4)		74 (11.3)
***Polypharmacy*** *(n*, *%)*			0.018	
No polypharmacy	58 (29.1)	146 (32.2)		204 (31.2)
Minor polypharmacy	85 (42.7)	225 (49.6)		310 (47.5)
Major polypharmacy	56 (28.1)	83 (18.3)		139 (21.3)
***Treatment adhering problem*** *(n*, *%)*			0.007	
Yes	35 (18.5)	128 (28.8)		163 (25.7)
No	154 (81.5)	317 (71.2)		471 (74.3)
***Use of medicinal plants*** *(n*, *%)*			0.579	
Yes	77 (38.9)	183 (41.2)		260 (40.5)
No	121 (61.1)	261 (58.8)		382 (59.5)

IQR: Interquartile range; n: number of interviewees per variable concerning the total number of interviewees; %: percentage of the variable about the total number of interviewees. Note: Missing data were not considered in the analyses.

**Table 3 ijerph-20-06143-t003:** Factors associated with self-medication in the population of Alegre, Espírito Santo.

	Multivariate Regression		
Variables	Adjusted PR	95%CI	*p*-Value
** *Age group* **			
≥60 years	1.00		
<60 years	1.13	1.01–1.26	0.036
** *Gender* **			
Men	1.00		
Women	1.19	1.04–1.37	0.010
** *Consumption of alcoholic beverages* **			
No	1.00		
Yes	1.13	1.01–1.25	0.028
** *Treatment adhering problem* **			
No	1.00		
Yes	1.15	1.04–1.28	0.006
** *Polypharmacy* **			
No polypharmacy	1.00		
Minor polypharmacy	0.98	0.88–1.10	0.785 *
Major polypharmacy	0.80	0.68–0.95	0.012

CI: Confidence interval; PR: Prevalence ratio. * Showed no statistical significance.

**Table 4 ijerph-20-06143-t004:** Medicines most used by the population practicing self-medication.

Item	Medicine	n (%)	ATC V	ATC II	ATC I	Classification	RENAME	Component	Self-Medication ^1^
		Total = 454							
1	Dipyrone	162 (35.7)	N02BB02	N02	N	OTC	Yes	Basic	Yes
2	Paracetamol	43 (9.5)	N02BE01	N02	N	OTC	Yes	Basic	Yes
3	Clonazepam	40 (8.8)	N03AE01	N03	N	Control	Yes	Basic	Partial
4	Omeprazole	29 (6.4)	A02BC01	A02	A	Prescribed	Yes	Basic	Partial
5	Dip + Orf + Caf	24 (5.3)	N02BB52	N02	N	OTC	No	-	Yes
6	AAS	19 (4.2)	B01AC06	B01	B	OTC	Yes	Basic	Yes
7	Nimesulide	16 (3.5)	M01AX17	M01	M	OTC	No	-	Yes
8	Levonorgestrel + ethinylestradiol	15 (3.3)	G03AA07	G03	G	Prescribed	Yes	Basic	Partial
9	Bromazepam	14 (3.1)	N05BA08	N05	N	Control	No	-	Partial
10	Alprazolam	13 (2.9)	N05BA12	N05	N	Control	No	-	Partial
11	Caf + Caris + Diclof + Paracet	12 (2.6)	N02BE51	N02	N	OTC	No	-	Yes
12	Fluoxetine	11 (2.4)	N06AB03	N06	N	Control	Yes	Basic	Partial
13	Phenyl + Paracet + Dexclor	8 (1.8)	N02BE51	N02	N	OTC	No	-	Yes
14	Citalopram	7 (1.5)	N06AB04	N06	N	Control	No	-	Partial

ASA: Acetylsalicylic acid; ATC: Anatomical Therapeutic Chemical Classification System; Caf + Caris + Diclof + Paracet: Caffeine + Carisoprodol + Diclofenac + Paracetamol; Dip + Orph + Caf: Dipyrone + Orphenadrine + Caffeine; OTC: Over-the-counter; Phenyl + Paracet + Dexclor: Phenylephrine + Paracetamol + Dexchlorpheniramine; RENAME: Brazilian National List of Essential Medicines. ^1^ Self-medication: **yes** = all interviewees reported using the drug for self-medication; **partial** = A portion of the interviewees, but not all, reported using the drug for self-medication.

**Table 5 ijerph-20-06143-t005:** Medicines most used by the population that does not practice self-medication.

Item	Medicine	n (%)	ATC V	ATC II	ATC I	Classification	RENAME	Component	Self-Medication ^1^
		Total = 454							
1	Losartan	97 (21.4)	C09CA01	C09	C	Prescribed	Yes	Basic	No
2	Hydrochlorothiazide	66 (14.5)	C03AA03	C03	C	Prescribed	Yes	Basic	No
3	Simvastatin	48 (10.6)	C10AA01	C10	C	Prescribed	Yes	Basic	No
4	Metformin	36 (7.9)	A10BA02	A10	A	Prescribed	Yes	Basic	No
5	Amlodipine	27 (5.9)	C08CA01	C08	C	Prescribed	Yes	Basic	No
6	Enalapril	25 (5.5)	C09AA02	C09	C	Prescribed	Yes	Basic	No
7	Atenolol	25 (5.5)	C07AB03	C07	C	Prescribed	Yes	Basic	No
8	Levothyroxine	24 (5.3)	H03AA01	H03	H	Prescribed	Yes	Basic	No
9	Glibenclamide	20 (4.4)	A10BB01	A10	A	Prescribed	Yes	Basic	No
10	Metoprolol	18 (4.0)	C07AB02	C07	C	Prescribed	Yes	Basic	No
11	Cholecalciferol	16 (3.5)	A11CC05	A11	A	Prescribed	No	-	No
12	Furosemide	15 (3.3)	C03CA01	C03	C	Prescribed	Yes	Basic	No
13	Captopril	14 (3.1)	C09AA01	C09	C	Prescribed	Yes	Basic	No
14	Propranolol	11 (2.4)	C07AA05	C07	C	Prescribed	Yes	Basic	No
15	Nifedipine	10 (2.2)	C08CA05	C08	C	Prescribed	Yes	Basic	No
16	Chlorthalidone	9 (2.0)	C03BA04	C03	C	Prescribed	No	-	No

ATC: Anatomical Therapeutic Chemical Classification System; RENAME: Brazilian National List of Essential Medicines. ^1^ Self-medication: **no** = none of the interviewees reported using the drug for self-medication.

## Data Availability

The datasets generated and analyzed during the current study are available from the corresponding author upon reasonable request.

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
