# Peer review of "Self-Medication during the COVID-19 Pandemic in Brazil: Findings and Implications to Promote the Rational Use of Medicines"

_ijerph, 2023, doi:10.3390/ijerph20126143_

Round 1

Reviewer 1 Report

 See the attached Word file. (Reviewer Comments)

Reviewer 2 Report

·         General comments:

·         The manuscript titled "Self-medication during the COVID-19 pandemic in Brazil: Findings and implications to promote the rational use of medicines" presents a study conducted to estimate the prevalence, profile, and associated factors of self-medication during the COVID-19 pandemic in Brazil. The topic is of relevance, considering the potential risks and implications of self-medication, especially during a global health crisis. The manuscript provides valuable insights into the prevalence and factors associated with self-medication in the study population. I would like to recommend it for publication. However, there are several areas where the manuscript can be improved to enhance clarity, strengthen the manuscript.

·         Major suggestions for revision:

·         Include specific numerical results and implications in the abstract: The abstract should provide concrete information about the prevalence of self-medication and the associated factors identified in the study. Including specific numbers or percentages would make the abstract more informative and impactful.

·         Incorporate more recent references in the introduction: While the introduction provides relevant background information, it would benefit from including more up-to-date references. This will ensure that the discussion reflects the most current understanding of self-medication and its implications.

·         The results section presents the findings in a clear and concise manner. The use of Poisson regression for identifying factors associated with self-medication is appropriate. However, it would be beneficial to include specific numerical results and effect sizes to provide a more comprehensive understanding of the associations. Additionally, discussing the limitations of the study, such as potential biases or generalizability issues, would further strengthen the interpretation of the results.

·         The discussion should start with a clear introduction that highlights the significance of self-medication and its implications for healthcare. It should provide a brief overview of the findings and their implications before delving into the comparison with previous studies.

·         The discussion would benefit from a more detailed explanation of the potential reasons behind the high prevalence of self-medication observed in the study. Factors such as accessibility of medications, lack of awareness about the risks, and economic considerations could be discussed.

·         It would be helpful to discuss the potential consequences of self-medication, such as delayed diagnosis, inappropriate use of medications, and the spread of diseases.

·         The discussion could explore the reasons for variations in the prevalence of self-medication observed in different studies and regions.

·         The limitations of the study should be addressed in more detail, including the cross-sectional design, recall bias, and the inability to capture the reasons behind self-medication.

·         Minor suggestions for revision:

·         Provide more details about the measurement of independent variables: In the methods section, it would be helpful to include information about how the independent variables were assessed or measured. This will enhance the transparency and replicability of the study.

·         In the first paragraph of the Results section, it would be helpful to mention the sampling method or strategy used to select the participants.

·         Consider providing more information about the data collection process, such as the timeframe and location of the interviews.

·         When reporting percentages in the tables, consider rounding to one decimal place for improved readability.

·         Consider rephrasing the sentence "All other factors analyzed failed to show statistical significance" to provide more clarity on the lack of statistical significance.

·         The authors should provide more specific details about the study, such as the methodology used, inclusion criteria, and data collection process.

·         When comparing the findings with previous studies, it would be helpful to provide more contextual information about those studies, such as their sample sizes, locations, and methodologies.

·         The authors should consider reorganizing the paragraph on factors associated with self-medication to provide a clearer and more concise overview. Each study's findings could be presented in a separate sentence or bullet point.

·         In the paragraph discussing the use of medications during the COVID-19 pandemic, it would be useful to elaborate on the potential reasons for increased self-medication during this period.

·         The discussion could provide more insight into the implications of the specific types of drugs commonly used for self-medication, such as analgesics and anti-inflammatories.

Addressing these suggestions will strengthen the manuscript and provide a more comprehensive and insightful analysis of self-medication during the COVID-19 pandemic in Brazil.

Reviewer 3 Report

This study is about the self-medication in COVID 19 time. I recommend publication of the study with minor modification as listed below:

-        In the introduction state along with aim and objectives of the study the importance of conducting such study and the importance of its result to the concerned authorities

-        In the methodology part you state two statistical programs were used in the analysis of data. Pease specify which analysis did you for each

-        From the result it can observed that the average age was high this should be explained in the discussion how it can affect the study result. The same applies foe preliminary school variable

-        I suggest changing the style of the table 4 so that it can be split into two sub tables one for self-medication and the other for non

-        In the conclusion give a small statement what work can be continued form this study

-        Double check the reference should consistent all over the text with either the full name or abbreviated name of the journal

-        Double check for grammar and typing mistakes
